# Population Structure and Resource Dynamics of Three Schizothoracinae Species in the Duoxiong Zangbo River Tributary of the Yarlung Zangbo River, Tibet: Threat Assessment and Conservation Insights

**DOI:** 10.3390/ani15162340

**Published:** 2025-08-10

**Authors:** Haoxiang Han, Lin Wang, Chi Zhang, Hongchi Li, Bo Ma

**Affiliations:** 1National Observation Experimental Station for Fishery Resources and Environment, Fuyuan, Scientific Observation and Experiment Station of Fishery Resources and Environment in Heilongjiang River Basin, Ministry of Agriculture and Rural Affairs, Heilongjiang River Fisheries Research Institute, Chinese Academy of Fishery Sciences, Harbin 150070, China; hanhaoxiang98@163.com; 2Institute of Fisheries Science, Tibet Academy of Agriculture and Animal Husbandry, Lhasa 850000, China; angels121@cafs.ac.cn (L.W.); zc0891@163.com (C.Z.); lihongchi19940702@163.com (H.L.); 3Chinese Academy of Fishery Science, Beijing 100141, China

**Keywords:** schizothoracinae, growth characteristic, exploitation rates, Duoxiong Zangbo River, Yarlung Zangbo River

## Abstract

*Schizothorax o*’*connori*, *Oxygymnocypris stewartii*, and other schizothoracinae fish species in the Duoxiong Zangbo River—a tributary of the Yarlung Zangbo River—are confronting severe threats, including population decline. Our study provides the first comprehensive assessment of key schizothoracinae species in these waters, revealing trends of early maturation, miniaturization, and accelerated growth. Current fishing pressure exceeds sustainable limits, indicating mild overexploitation. These findings serve as an urgent ecological warning, prompting us to propose targeted management strategies to support the conservation and restoration of these fish populations.

## 1. Introduction

The Tibetan Plateau serves as a vital ecological barrier in global geography [1,2] and represents one of the world’s highest and most complex plateaus [3,4]. Often referred to as the “Third Pole of the Earth” [1], its distinctive alpine climate system plays a pivotal role in global water vapor cycles and biodiversity maintenance [5]. However, Tibetan Plateau ecosystems are notably fragile and highly sensitive to environmental changes [6,7]. Under the dual pressures of climate change and human activities [4,8,9], significant ecological responses have emerged, including hydrological cycle disruption [10] and persistent declines in river runoff [11,12,13].

As a crucial water system on the southern Tibetan Plateau, the Yarlung Zangbo River originates from the Jema Yangzong Glacier [14]. With a total length of 2057 km, the river forms part of the Ganges–Brahmaputra River system. Its main channel and tributaries collectively form an intricate hydrological network [15], recognized as one of Asia’s most ecologically significant watersheds [16,17]. The region’s distinctive high-altitude geomorphology and climatic conditions [18,19,20] have fostered specialized aquatic organisms adapted to extreme environments [21], exhibiting unique evolutionary trajectories [22]. Notably, schizothoracinae have undergone exceptional evolutionary diversification here, developing distinct niche differentiation. Their survival strategies markedly differ from lowland fishes, including evolutionary adaptations such as slow growth [23,24], extended life cycles [25], delayed sexual maturity [26,27,28,29], and reduced fecundity [27,30]. These life-history traits result in limited resilience to environmental disturbances [31]. Furthermore, the plateau’s fragile aquatic ecosystem [32] faces compounded environmental stressors [33,34] that are highly sensitive to both climatic variability and anthropogenic impacts [4,8,35], posing severe threats to these species’ survival [31]. Consequently, research on the Yarlung Zangbo River Basin holds substantial ecological value and strategic importance.

Tributaries constitute a vital component of watershed systems, serving as primary recharge sources and playing a crucial role in maintaining ecological balance [3]. Among the Yarlung Zangbo River’s tributaries, the Lhasa and Nianchu Rivers feature open channels, elevated water temperatures [36], and gentle bank slopes [37]. However, these rivers flow through major urban centers (Lhasa and Nyingchi) [38] and face increasing pressures from water system fragmentation caused by intensive fishing, graded hydropower development, and invasive species. These anthropogenic activities have substantially reduced ecological niches and breeding habitats for indigenous fish species. Studies indicate that persistent external pressures have led to overexploitation of *Schizopygopsis younghusbandi* populations in the Nianchu, Lhasa, and Niyang rivers, resulting in population miniaturization and juvenilization [39]. In contrast, the Duoxiong Zangbo River tributary provides abundant habitats and biomass for indigenous fish due to its high altitude, low temperatures, sparse vegetation cover, remote location [29], and minimal human disturbance [36]. Historical records show stable populations of endemic schizothoracinae such as *Oxygymnocypris stewartia* (Lloyd, 1908), *Ptychobarbus dipogon* (Regan, 1905), and *Schizothorax o*’*connori* (Lloyd, 1908) [29,40]. However, recent regional economic development has spurred water conservancy projects [19], intensified fishing, and increased biological invasions [41], exceeding the aquatic ecosystem’s resilience threshold and causing a dramatic decline in schizothoracinae resources. These external pressures will inevitably impact the ecosystem of the Duoxiong Zangbo River waters. However, the extent of their effects on local fish stocks remains unclear. Thus, it is imperative to investigate the population structure, growth characteristics, resource dynamics, and degree of external disturbance in this region.

Fish community structure serves as a crucial biological indicator of aquatic ecosystem health [42,43]. Environmental pressures are increasingly altering community structures, jeopardizing fish survival and reproduction [44,45,46,47] and contributing to the degradation of river fishery function [45,48]. Consequently, establishing adaptive management mechanisms based on ecological carrying capacity has become imperative [29]. The effectiveness of such mechanisms relies on comprehensive understanding of key species’ population dynamics and their response to environmental stressors. Age structure, growth parameters, and mortality rates constitute fundamental elements for analyzing population dynamics and predicting resource trends [49,50,51], providing essential data for management strategies. The unique schizothoracinae fish resources in Yarlung Zangbo River tributaries, particularly *S. o’connori*, *O. stewartii*, and *P. dipogon*, hold particular ecological and resource management significance [52]. This study hypothesizes significant alterations in these species’ population structures and growth characteristics, with observed increases in total mortality primarily attributable to intensive anthropogenic activities (especially fishing pressure) that substantially outweigh natural mortality. To test these hypotheses and establish baseline ecological data, we conducted a two-year fishery survey in the Duoxiong Zangbo River, analyzing the age structure, growth patterns, and resource status of the target species (*S. o’connori*, *O. stewartii*, and *P. dipogon*). Furthermore, we developed a population dynamics model to quantitatively evaluate the relative impacts of natural versus anthropogenic disturbances (particularly fishing) on total mortality and its components (fishing mortality (*F*) and natural mortality (*M*)). These findings aim to elucidate schizothoracinae fish population responses to environmental stress while informing conservation strategies and adaptive fisheries management in the Duoxiong Zangbo and Yarlung Zangbo River basins.

## 2. Materials and Methods

### 2.1. Sample Collection and Processing

Six sampling sites were established along Duoxiong Zangbo River, a tributary of the upper Yarlung Zangbo River (Figure 1). Sampling points were established following standardized criteria that account for habitat variation, anthropogenic disturbance gradients, and tributary confluences. To ensure spatial independence, adjacent sampling points were positioned at minimum intervals of 20 km. Four field surveys were conducted between 2023 and 2024 during two seasons: summer (June–July) and autumn (September–October). Sampling activities were authorized by the fisheries authorities of the Tibet Autonomous Region. Standardized fishing gear combinations were employed at each site, including eight gillnets (two each with mesh sizes of 2–5 cm, 20 m long, and 1 m high) and two ground cages (6 m long, 40 cm wide, and 40 cm high). We conducted passive fishing for 12 consecutive hours from 18:00 to 06:00 the following day.

Each captured individual was measured for total length, body length (±1 mm), and body weight (±0.1 g). Following dissection for sex identification, otoliths were extracted and preserved in anhydrous ethanol to remove connective tissue, then air-dried at ambient temperature (25 ± 2 °C) before storage in 1.5 mL centrifuge tubes. In the laboratory, otoliths were mounted on slides using colorless nail polish; air-dried; and sequentially polished using 800-grit#, 1000-grit#, and 3000-grit# abrasive papers under real-time stereomicroscope observation. The polishing process continued until the central growth whorl patterns became clearly visible (repeated on both sides if necessary). Annual growth bands (dark and transparent bands) were identified under incident light, with the dark band serving as the annual marker for counting and microimaging.

### 2.2. Data Statistics and Analysis Methods

#### 2.2.1. Body Length–Body Weight Relationship

The body length–weight relationship was modeled using a power function [53], and analysis of covariance (ANCOVA) was applied to test for significant differences in this relationship between males and females. The equations were derived as follows:*W* = a *L^b^*
where *W* represents body weight (g), *L* denotes body length (mm), and *a* and *b* are constant.

To assess the statistical significance of the difference between parameter “*b*” and value 3, the *t*-test formula (*t* = $\frac{SD(L)}{SD(w)}$×$\frac{\left|b-3\right|}{\sqrt{1-r2}}$×$\sqrt{n-2}$) was employed.

#### 2.2.2. Growth Equation, Growth Rate Equation, and Growth Acceleration Equation

The von Bertalanffy growth equation was employed to model body length growth [54], while the body mass growth equation was derived from the length–mass relationship. The first and second derivatives of these growth equations were computed to obtain the corresponding growth rate and acceleration equations. Subsequently, the inflection point age, critical age, and growth performance index were calculated as follows:Body length growth equation: *L*_t_ = *L*_∞_ (1 − *e*^−*k(t−t*^_0_^)^);Body weight growth equation: *W_t_* = *W*_∞_ (1 *− e^−k(t−t^*_0_^)^)*^b^*;Body length growth rate equation: *dL/dt* = *L_∞_ k e^−k(t−t^*_0_^)^;Body weight growth rate equation: *dW/dt* = *b W_∞_ k e^−k(t−^^t^*_0_^)^ (1 *− e^−k(t−^^t^*_0_*^)^*)*^b−^*^1^;Body length growth acceleration equation: *d*^2^*L/dt*^2^ = *−L_∞_k*^2^*e^−k^*^(*t*−*t*^_0_^)^;Body weight Growth Acceleration Equation: *d*^2^*W*/*dt*^2^ = *bW*_∞_*k*^2^*e − k* (*t − t*_0_) (1 − *e^−k^*^(*t−t*^_0_^)^)*^b−^*^2^ (*be^−k(t−t^*_0_^)^ − 1);Inflection point age: *t_i_* = ln *b*/*k + t*_0_;Critical age: *t_c_* = [*Kt*_0_ − ln*M* + ln (*bK + M*)]/*K*;Growth performance index: φ = lg*k* + 2 lg*L*_∞_;
where *t* represents the age; *t*_0_, *t_i_*, and *t_c_* represent the theoretical starting age, inflection point age, and critical age, respectively; *L_t_* and *W_t_* represent the body length (mm) and weight (g) at age *t*; *L*_∞_ represents the asymptotic body length (mm); *W*_∞_ represents the asymptotic body weight (g); and *k* represents the growth coefficient.

#### 2.2.3. Mortality Characteristics and Exploitation Rate

The total mortality coefficient (*Z*) was calculated using the formula proposed by Beverton–Holt. The natural mortality coefficient (*M*) was estimated using Pauly’s formula [55,56,57]. The fishing mortality coefficient (*F*) was derived as the total mortality coefficient (*Z*) minus the natural mortality coefficient (*M*). The population exploitation rate (*E*) was calculated as the ratio of the fishing mortality coefficient (*F*) to the total mortality coefficient (*Z*), as follows:*Z* = *K* (*L*_∞_ − *L_mean_*)/(*L_mean_* − *L_c_*)ln*M* = −0.0066 − 0.279 ln*L*_∞_ + 0.6543 ln*K* + 0.4634 ln*T**F* = *Z* − *M**E* = *F*/*Z*
where *K* represents the growth rate, *L*_∞_ represents the asymptotic body length (mm), *L_mean_* represents the average body length of the sample (mm), *L_c_* represents the starting body length (mm), and *T* represents the annual average water temperature (°C) of the habitat for the studied fish species.

#### 2.2.4. Relative Yield per Recruit and Biomass per Recruit

Resource utilization by schizothoracinae was assessed using the Beverton–Holt dynamic pool model, specifically through relative yield per recruit (Y′/R) and relative biomass per recruit (B′/R) curves. The relevant model equations are expressed as follows [56]:*Y*′/*R* = *E* × (1 − *L_c_*/*L*_∞_) ^M/*k*^ [1 − 3(1 − *L_c_*/*L*_∞_)/(1 + *k/*Z) + 3 (1 − *L_c_*/*L*_∞_)^2^/(1 + 2 *k/*Z) − (1 − *L_c_*/*L*_∞_)^3^/(1 + 3 *k/*Z)]*B*′/*R* = (*Y*′/*R*)/(*Z* − *M*)
where *E* represents the exploitation rate, *L*_∞_ (mm) represents the asymptotic body length, *L_c_* (mm) represents the length of the starting body, *M* represents the natural mortality factor, *Z* represents the total mortality coefficient, and *k* represents the growth factor.

## 3. Results

### 3.1. Group Structure

During this survey, we collected 51 *S. o*’*connori*, 45 *O. stewartii*, and 97 *P. dipogon* specimens from DuoXiong Zangbo River (Appendix A). The schizothoracinae specimens exhibited body lengths ranging from 23.02 to 460.00 mm and body weights ranging from 0.20 to 1394.30 g. The male-to-female ratios were 0.57:1 for *S. o*’*connori*, 1:1 for *O. stewartii*, and 0.79:1 for *P. dipogon*. Age distributions spanned 0–13 years for *S. o*’*connori*, 0–16 years for *O. stewartii*, and 0–14 years for *P. dipogon* (Table 1, Figure 2).

### 3.2. Body Length–Body Weight Relationship

Analysis of covariance revealed no significant sexual dimorphism in the length–weight relationship for any of the three schizothoracinae species: *O. stewarti* (*F*_(1,37)_ = 0.901, *p* = 0.349), *P. dipogon* (*F*_(1,92)_ = 1.409, *p* = 0.259), and *S. o*’*connori* (*F*_(7,37)_ = 2.178, *p* = 0.059). Consequently, a unified length–mass relationship was established for each species (Figure 3).

Independent-sample *t*-tests were conducted to compare the growth coefficient (b) of the length–weight regression equations among the three fish populations with the theoretical value of 3. The results showed no statistically significant differences for *O. stewarti* (*t* = 0.59, *p* > 0.05, *df* = 44), *P. dipogon* (*t* = 0.88, *p* > 0.05, *df* = 96), or *S. o’connori* (*t* = 1.07, *p* > 0.05, *df* = 50), as all t-values were below their respective critical values (t_0.05_ = 2.021, 1.987, and 2.009). This suggests that the growth patterns of these three fish species are consistent with isometric growth (b ≈ 3) (Table 2).

### 3.3. Growth Equation, Growth Velocity Equation, and Growth Acceleration Equation

This study employed empirical measurements to successfully model the growth patterns of three schizothoracinae fish species using the von Bertalanffy growth function (VBGF) (Figure 4 and Figure 5, Table 2). Model derivation yielded equations characterizing both growth velocity (first derivative) and acceleration (second derivative). Analytical results demonstrated that while the three species shared similar overall growth trends, they exhibited significant differences in dynamic details. Analysis of length growth revealed continuously decreasing velocity and acceleration throughout ontogeny, with persistent negative acceleration indicating a monotonically decelerating process devoid of inflection points. In contrast, mass growth displayed distinct dynamics, featuring a characteristic inflection point (Figure 5). Pre-inflection growth showed annually increasing velocity (positive acceleration), marking a phase of progressive weight gain. Maximum growth velocity occurred at the inflection age, followed by declining growth rates (negative acceleration) with diminishing mass accumulation efficiency. Key growth parameters were quantitatively characterized (Table 3). *S. o*’*connori* exhibited significantly greater values for both mass growth inflection age (maximum growth velocity) and critical age (theoretical growth cessation) compared to the other two species. However, growth performance indices (φ’) showed no significant interspecific variation (*p* > 0.05), indicating comparable efficiencies in energy conversion to somatic growth across all three taxa.

### 3.4. Mortality Characteristics and Exploitation Rate

Exploitation rates (*E*) of the three schizothoracinae species ranged from 0.547 to 0.758. *O. stewartii* showed the highest exploitation rate (0.758), followed by *P. dipogon* (0.711), while *S. o*’*connori* showed the lowest (0.547). All rates were below *E*-max (Table 4).

### 3.5. Relative Yield per Recruit and Biomass per Recruit

The relationship between the relative yield per recruit (*Y′/R*) and relative biomass per recruit (*B′/R*) for the three schizothoracinae species as a function of the exploitation rate (*E*) is presented in Figure 6. *Y′/R* peaked when the exploitation rate reached its maximum value (*S. o*’*connori*: 0.579; *O. stewartii*: 0.882; *P. dipogon*: 0.884), declining sightly as the rate decreased to *E*-50. At maximum exploitation (*E*-max), relative biomass was below 25% for all species but reached 50% at *E*-50. Contour analysis of *Y′/R* versus *E* and *L_c_* revealed that, under the current exploitation rate, *Y′/R* increased with Lc, attaining its maximum at *L_c_*/*L*_∞_ = 0.6 (Figure 7). The corresponding optimal catch lengths at this point were 355 mm for *S. o*’*connori*, 305 mm for *O. stewartii*, and 309 mm for *P. dipogon*.

## 4. Discussion

### 4.1. Population Structure and Growth Characteristics

The growth process of fish is jointly regulated by genetic and environmental factors [58,59]. Notably, conspecific fish often exhibit significant growth variations across different distribution areas or environmental conditions. Our analysis of growth patterns in fish from the Duoxiong Zangbo River waterbody revealed that the allometric coefficient (*b*) values for *S. o’connori* (*b* = 2.9375 [2.886, 2.989]), *O. stewarti* (*b* = 2.9576 [2.902, 3.014]), and *P. dipogon* (*b* = 2.9459 [2.875, 3.017]) all include the theoretical isometric value of 3 within their 95% confidence intervals. This strongly supports an isometric growth model, indicating that these species maintain geometric similarity in body shape during development, with minimal interspecific differences. The slightly lower point estimates suggest a potential weak negative allometric growth trend, likely influenced by environmental conditions in the Duoxiong Zangbo River [60,61]. Compared to the middle reaches of the Niyang and Lhasa Rivers [37], this high-altitude waterbody exhibits lower temperatures and reduced food availability [34,36,38], which may lead to preferential energy allocation toward longitudinal growth rather than mass accumulation [61]. However, since all confidence intervals encompass a value of 3, this minor deviation requires further investigation to determine its biological significance.

Fish growth parameters serve as fundamental data for the study of biological characteristics, population structure, and resource assessment and are critical components in constructing fishery resource models. These parameters directly influence population biomass and distribution patterns [62,63,64,65]. Among them, the growth coefficient (*k*) is a key indicator of fish population growth potential [66]. Branstetter [67] classified fish growth rates into three types based on *k*-values: slow (*k* = 0.05–0.10), medium (*k* = 0.10–0.20), and fast (*k* = 0.20–0.50). In the Duoxiong Zangbo River, *O. stewartii* (*k* = 0.122) and *P. dipogon* (*k* = 0.118) exhibit medium growth rates, while *S. o*’*connori* (*k* = 0.098) falls into the slow-growth category, suggesting limited population growth potential. Comparative analysis reveals that most *S. o*’*connori* subfamily members have *k* values around 0.1, indicating a generally slow growth rate. Consequently, the populations of this subfamily may struggle to recover following resource depletion.

Comparative temporal analysis revealed that the growth coefficient (k) of *S. o*’*connori* in the Duoxiong Zangbo River watershed was significantly higher than that of the 2008–2009 Yarlung Zangbo River mainstem population (*k*_male_ = 0.095; *k*_female_ = 0.081) [68]. Similarly, the *k* value of *P. dipogon* in this watershed exceeded that of the 2019 Yarlung Zangbo River mainstem population (*k*_male_ = 0.069; *k*_female_ = 0.048) [69]. This acceleration stems from two primary factors: habitat environmental changes [70] and fishing pressure [71]. Specifically, rising water temperatures due to climate warming have enhanced fish metabolism and growth rates [9,35], while intensive fishing has prompted schizothoracinae populations to adapt through phenotypic plasticity [62], manifesting as compensatory growth to maintain population viability [72,73]. Regarding environmental adaptations, fish often adjust their population age structure. Our comparison of historical data for three schizothoracinae species in the Yarlung Zangbo River Basin showed maximum ages of 50 years for *S. o*’*connori* [68] and 47 years for *P. dipogon* [74], whereas the maximum age in our study was only 16 years. This substantial discrepancy suggests significant alterations in age structure and lifespan, likely attributable to environmental changes and increased fishing pressure. The elevated growth rates observed in these schizothoracinae species directly reflect intense fishing pressure [75], with these adaptive responses causing marked shifts in life-history parameters. Collectively, these changes indicate a trend toward younger and smaller population structures.

The growth coefficient (*k*) is sensitive to variations in population age structure, whereas the growth performance index (φ), which integrates information from both k and *L_∞_*, is commonly used to evaluate the reliability of growth parameter estimates. Notably, φ values tend to be similar among closely related species [76]. In this study, we employed φ values to assess the accuracy of growth parameters for *S. o*’*connori*, *O. stewartii*, and *P. dipogon* in the Duoxiong Zangbo River watershed and to examine their interspecific growth variation. The φ’ values for these three schizothoracinae species ranged narrowly from 4.50 to 4.53, suggesting robust and reliable growth parameter estimates. Comparative analysis with other Tibetan Plateau schizothoracinae populations revealed that our φ’ values fell within the median range (4.37–4.75) of previously reported values [77,78,79,80,81,82,83]. This indicates that the growth performance of these species in Duoxiong Zangbo River is intermediate among highland schizothoracinae. Their life-history strategies appear primarily constrained by phylogenetic factors, maintaining energy allocation patterns consistent with their sympatric relatives. The observed growth patterns likely reflect phenotypic plasticity in response to local environmental conditions rather than profound genetic adaptation. These findings provide crucial quantitative support for the development of watershed-specific aquatic resource restoration strategies based on ecological characteristics.

### 4.2. Resource Dynamics and Exploration Rate

Fish mortality is a critical driver of population dynamics, directly influencing the magnitude of population decline. In this study, we evaluated mortality parameters for *S. o*’*connori*, *O. stewartii*, and *P. dipogon* in the Duoxiong Zangbo River watershed using established reliability criteria: the growth equation validity threshold (*e^−k^* < 1) [84], the natural mortality confidence range (*M/K* = 1.5–2.5), and the mortality-type discrimination threshold (*Z/K* > 3) [84]. The *e^−k^* values (0.89–0.91) confirmed the validity of the growth model, while M/K ratios (2.07–2.14) aligned with theoretical expectations for natural mortality. These results support the reliability of our growth parameter (*k*, *L_∞_*) estimates and natural mortality (M) calculations. Notably, the *Z/K* ratios (4.72–8.53) for all three species substantially exceeded the discrimination threshold (*Z/K* > 3), indicating that total mortality (*Z*) is predominantly driven by anthropogenic fishing pressure. This suggests severe overexploitation of these schizothoracinae populations, necessitating immediate fisheries management interventions to mitigate further resource depletion. This study primarily aims to elucidate the driving mechanisms of overfishing through mortality structure analysis (*Z/K* >> 3). Although direct reports on natural mortality rates (*M*) of these endemic fish species are scarce in the literature, the estimated M values in this study were rigorously validated against biological criteria (*M/K* = 2.07–2.14 ∈ 1.5–2.5), thereby providing robust support for our conclusions. The determination of population-critical thresholds (e.g., minimum viable population, MVP) requires long-term integration of population viability analysis (PVA) data, which constitutes a distinct research focus. The substantial exceedance of the *Z/K* threshold (>3) observed here (4.72–8.53) provides conclusive evidence of overfishing, underscoring the need for immediate management interventions. Future studies should build upon these findings to quantify population recovery targets more precisely.

The definition of overexploitation thresholds for fish resources remains debated. While Gulland [85] proposed an exploitation rate threshold of 0.5 to identify overexploited populations, Mehanna [86] suggested that rates below *E*-max indicate sustainable exploitation. In this study, the exploitation rates of *S. o*’*connori* (0.547), *O. stewartii* (0.758), and *P. dipogon* (0.711) exceeded Gulland’s threshold but remained below Mehanna’s proposed Emax for riverine systems, suggesting moderate exploitation. This intermediate status may reflect the remote location of the Duoxiong Zangbo River watershed, which lacks major urban centers and experiences limited anthropogenic pressure. Additionally, traditional Tibetan religious practices may contribute to informal fish conservation, distinguishing this region from more heavily exploited tributaries in the middle and lower Yarlung Zangbo River basin [39].

Analysis using the Beverton-Holt dynamic synthesis model identified suboptimal exploitation patterns in the stock resources of three schizothoracinae species (*S. o*’*connori*, *O. stewartii*, and *P. dipogon*). Based on integrated exploitation rate analysis, current fishing practices exhibit excessive intensity and small size at capture. Thus, management optimization requires an increase in the minimum capture size and reduced fishing pressure. Notably, adjusting size limits may offer greater conservation benefits than simply decreasing fishing intensity [87,88]. We propose three science-based methods for determining optimal size at capture: the inflection point length for maximum commercial yield, the critical length at critical age for generational biomass conservation, and the length corresponding to peak yield per recruit in the Beverton–Holt model. By averaging these metrics, we derived optimal opening lengths of 353 mm (*S. o*’*connori*), 313 mm (*O. stewartii*), and 317 mm (*P. dipogon*). Implementing these size restrictions would enhance stock recruitment while maintaining sustainable harvests, allowing individuals to realize their growth potential, preventing growth overfishing, and ensuring long-term resource sustainability [87].

### 4.3. Conservation Measures

Based on analyses of growth characteristics in *S. o*’*connori*, *O. stewartii*, and *P. dipogon*, coupled with assessments of aquatic ecosystem vulnerability on the Qinghai-Tibet Plateau, this study proposes four core conservation strategies: (1) Protected area establishment: Designate no-fishing zones in critical habitats (spawning and feeding grounds) [89,90,91,92,93], complemented by artificial breeding and stock enhancement programs [94,95]. This integrated approach aims to rebuild natural populations and reverse resource depletion trends. (2) Stocking regulation: Develop science-based release protocols with rigorous approval and monitoring systems [93,96], prioritizing prevention of non-native species introductions to maintain ecosystem integrity. (3) Fisheries management: Implement catch quotas and seasonal closures [87,92,97,98,99,100,101,102] to protect reproductive cycles and growth phases while reducing habitat degradation and illegal harvests. (4) Ecological monitoring: Create a comprehensive aquatic monitoring network [95] for systematic tracking of hydrological conditions and fish community dynamics, enabling adaptive management.

## 5. Conclusions

This study presents a comprehensive evaluation of resource status, growth characteristics, and exploitation intensity for three schizothoracine fish species (*S. oconnori*, *O. stewarti*, and *P. dipogon*) in the Duoxiong Zangbo River basin. Key findings indicate the following: (1) Growth parameter analysis (asymptotic length *L_∞_* = 507.577–591.233 mm; growth coefficient *k* = 0.098–0.122), combined with exploitation rate assessment (current values = 0.547–0.758), reveals all three populations are moderately exploited. While current exploitation rates exceed the general sustainability threshold (0.5), none has reached the maximum sustainable rate (*E*-max) predicted by the Beverton–Holt model. (2) Beverton–Holt dynamic modeling identifies critical management deficiencies, particularly excessive fishing pressure and suboptimal minimum size limits, which threaten long-term population stability if unaddressed. (3) Considering species-specific growth parameters (inflection age and critical age) and sustainable yield optimization, we recommend raising minimum size limits to 355 mm (*S. oconnori*), 305 mm (*O. stewarti*), and 309 mm (*P. dipogon*). These adjustments would protect pre-reproductive individuals, ensure recruitment, and balance ecological sustainability with economic benefits.

## Figures and Tables

**Figure 1 animals-15-02340-f001:**
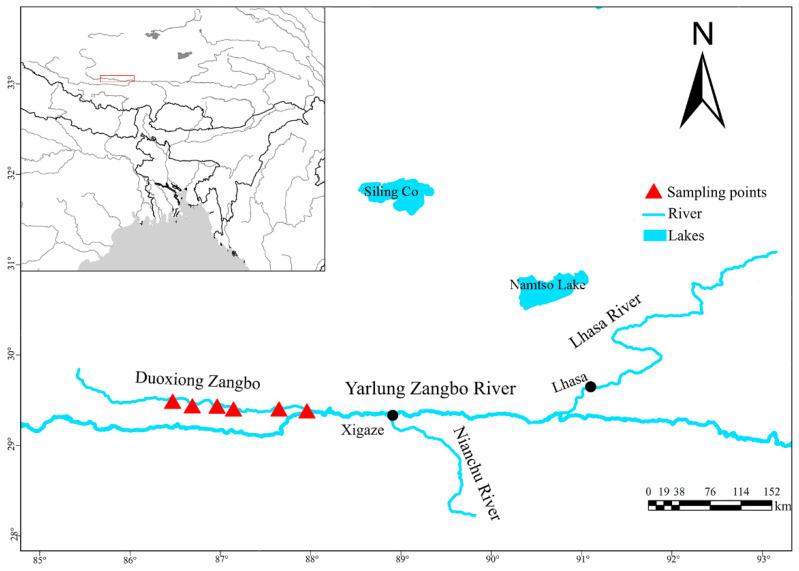
Sampling site.

**Figure 2 animals-15-02340-f002:**
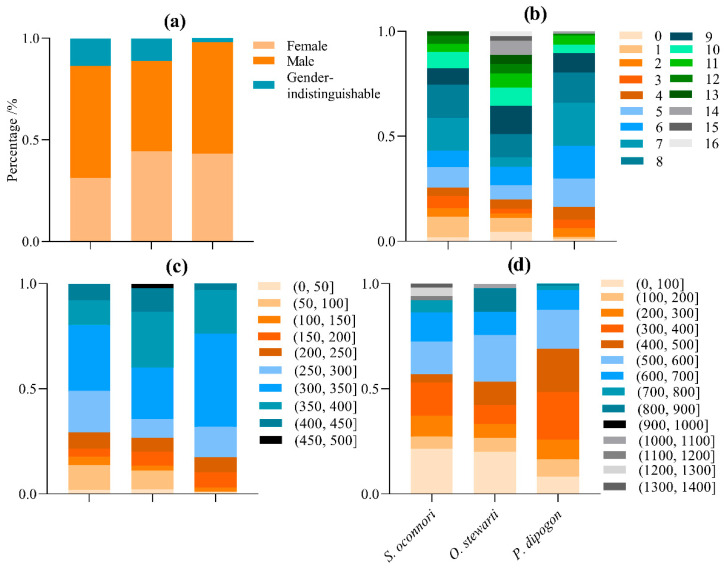
The population structure of three schizothoracinae species. (**a**) Sex proportion; (**b**) age proportion; (**c**) proportion of the body length group; (**d**) proportion of the body weight group.

**Figure 3 animals-15-02340-f003:**
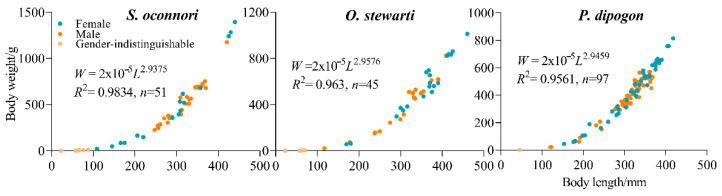
Relationship between body length and body weight of three schizothoracinae species.

**Figure 4 animals-15-02340-f004:**
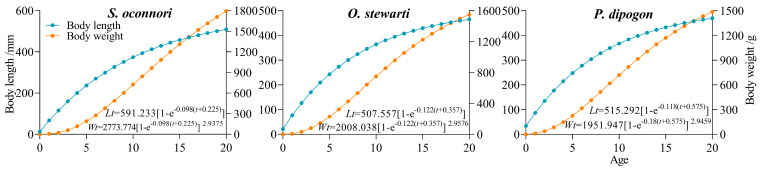
Growth equation of three schizothoracinae species.

**Figure 5 animals-15-02340-f005:**
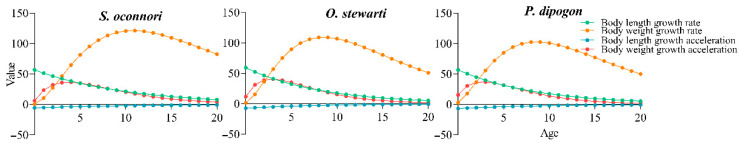
Growth rate and growth acceleration of three schizothoracinae species.

**Figure 6 animals-15-02340-f006:**
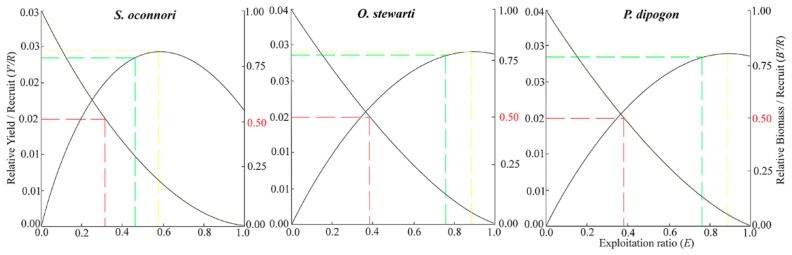
Two-dimensional analysis of *Y*′/*R* and *B*′/*R* of three schizothoracinae species.

**Figure 7 animals-15-02340-f007:**
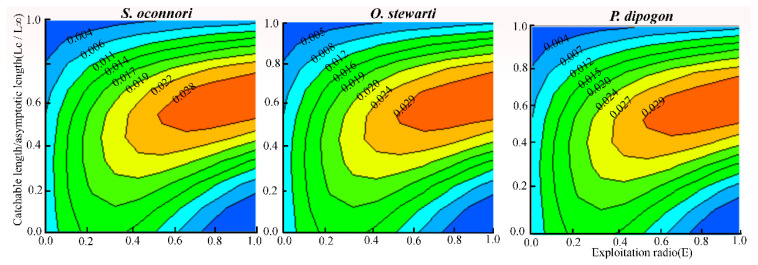
Relative yield per recruit (*Y*′/*R*) of three schizothoracinae species in relation to exploitation rate and catchable length.

**Table 1 animals-15-02340-t001:** The body lengths and body weights of three schizothoracinae species.

Species Name	Number	Number		Body Length/mm		Body Weight/g	
		Female	Male	Range	Mean	Range	Mean
*S. o*’*connori*	51	16	28	23.02~440.00	268.77	0.3~1394.3	430.50
*O. stewartii*	45	20	20	23.02~460.00	298.37	0.2~1013.0	425.33
*P. dipogon*	97	42	53	45.18~418.00	307.01	1.2~814.3	399.90

**Table 2 animals-15-02340-t002:** Confidence intervals for relevant growth parameters of three schizothoracinae species.

Species	*b* CL 95%	*L_∞_* CL 95%	*k* CL 95%	*to* CL 95%
*S. o*’*connori*	2.886, 2.989	511.329, 671.136	0.075, 0.121	−0.504, 0.054
*O. stewarti*	2.902, 3.014	474.529, 540.586	0.104, 0.139	−0.609, −0.105
*P. dipogon*	2.875, 3.017	477.818, 552.766	0.100, 0.137	−0.895, −0.255

**Table 3 animals-15-02340-t003:** The inflection point age, critical age, and growth performance index of three schizothoracinae species.

Species Name	Inflection Point Age	Critical Age	Growth Performance Index
*S. o*’*connori*	10.77	8.60	4.53
*O. stewartii*	8.53	6.92	4.50
*P. dipogon*	8.58	6.89	4.50

**Table 4 animals-15-02340-t004:** Mortality coefficient and exploitation rate of three schizothoracinae species.

Species Name	Total Mortality (*Z*)	Natural Mortality (*M*)	Exploitation Rate (*E*)	*E*-Max
*S. o’connori*	0.463	0.210	0.547	0.579
*O. stewartii*	1.041	0.252	0.758	0.882
*P. dipogon*	0.85	0.246	0.711	0.884

## Data Availability

The data that support the findings of this study are available from the corresponding author upon reasonable request.

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
