# Peer review of "Population Structure and Resource Dynamics of Three Schizothoracinae Species in the Duoxiong Zangbo River Tributary of the Yarlung Zangbo River, Tibet: Threat Assessment and Conservation Insights"

_animals, 2025, doi:10.3390/ani15162340_

Round 1
Reviewer 1 Report
Comments and Suggestions for Authors
Dear Authors,
The manuscript touches upon a practical topic. Endemic species in exploitation should always be under strict control. However, the correct formulation of the manuscript title and hypothesis should reveal the meaning of the research work. Unfortunately, this is not the case yet. The authors have not sufficiently described the species composition of fish at the collection points. This is important. They have not described possible competitive relationships with other animals. This should also be disclosed on the pages of the manuscript. The text of the manuscript is mixed up in different chapters and should be moved to the appropriate chapters. In many methodological aspects, the authors omit important information. It should be added. The comparative part in the discussion needs to be expanded and additional sources of literature on other fish species should be cited. The focus should be on writing the conclusion of the manuscript based on the analysis of the research results. The authors obtained interesting results, presented them and they should be disclosed in the conclusions of the manuscript from a practical point of view for the future of 5, 10, 50, 100 years. After eliminating all the comments, the manuscript can be reviewed again.

Reviewer 2 Report
Comments and Suggestions for Authors
In the length-weight relationship, they indicate that growth is allometric when the value of b differs significantly from 3, but they do not say what statistical test they used to evaluate the significance of this value.
In the growth equation in length, the letter e is different from the rest of the equations. I suggest standardizing the letter.
The weight-growth equation requires a curvature parameter. In the Von Bertalanffy model, this parameter is fixed at 3, but it can be left free when fitting.
Figure 2, in panels b, c and d are not conventional and together with the scale of the Y axis it is difficult to interpret
Subtopic 3.2 of results does not present the confidence interval for the value of b
In section 3.3, I notice that they forced the Bertalanffy model by weight with the LWR value of b. Therefore, it is more important to be stricter in obtaining the parameter b.
In the discussion section, they start with the value of b and, for example, say that a value of 2.95 is below 3 compared to other sites where it is above 3 and mention a value of 3.056. It is necessary to know the confidence interval both in this study and against those with which it is being compared. The confidence interval may reveal that no value is significantly different from 3.
The growth parameter values also do not have a confidence interval to make a stronger discussion.
Reviewer 3 Report
Comments and Suggestions for Authors
See comments in the attached file

Round 2
Reviewer 1 Report
Comments and Suggestions for Authors
Dear Authors,
I am satisfied with the revision of the manuscript. The manuscript has been corrected and additional data have been included. This study presents a comprehensive evaluation of resource status, growth characteristics, and exploitation intensity for three schizothoracine fish species (S. oconnori, O. stewarti, and P. dipogon) in the Duoxiong Zangbo River basin. The review of the study results was chosen appropriately, as were the statistical methods used for its analysis. The article takes into account the comments on the methodology. The analysis and conclusion for each chapter are sufficient and do not raise objections. References to sources of literature have been adjusted. The results of previous studies by other authors have been taken into account. I recommend it for the journal Animals.
Reviewer 3 Report
Comments and Suggestions for Authors
I believe that all corrections made were adequately addressed, so I believe the corrected version can be accepted for publication.